# GalleryGPT: Analyzing Paintings with Large Multimodal Models

Yi Bin
Tongji University
Shanghai, China
National University of Singapore
Singapore
yi.bin@hotmail.com

Wenhao Shi
University of Electronic Science and
Technology of China
Chengdu, China
shiwenhao16@gmail.com

Yujuan Ding*
The Hong Kong Polytechnic University
Hong Kong SAR, China
dingyujuan385@gmail.com

Zhiqiang Hu
Singapore University of Technology and Design
Singapore
zhiqiang_hu@mymail.sutd.edu.sg

Zheng Wang
Tongji University
Shanghai, China
zh_wang@hotmail.com

Yang Yang
University of Electronic Science and
Technology of China
Chengdu, China
yang.yang@uestc.edu.cn

See-Kiong Ng
National University of Singapore
Singapore
seekiong@nus.edu.sg

Heng Tao Shen
Tongji University
Shanghai, China
University of Electronic Science and
Technology of China
Chengdu, China
shenhengtao@hotmail.com

## ABSTRACT

Artwork analysis is an important and fundamental skill for art appreciation, which could enrich personal aesthetic sensibility and facilitate the critical thinking ability. Understanding artworks is challenging due to its subjective nature, diverse interpretations, and complex visual elements, requiring expertise in art history, cultural background, and aesthetic theory. However, limited by the data collection and model ability, previous works for automatically analyzing artworks mainly focus on classification, retrieval, and other simple tasks, which is far from the goal of AI. To facilitate the research progress, in this paper, we step further to compose comprehensive analysis inspired by the remarkable perception and generation ability of large multimodal models. Specifically, we first propose a task of composing paragraph analysis for artworks, *i.e.*, painting in this paper, only focusing on visual characteristics to formulate more comprehensive understanding of artworks. To support the research on formal analysis, we collect a large dataset Painting-Form, with about 19k painting images and 50k analysis paragraphs. We further introduce a superior large multimodal model for painting analysis composing, dubbed GalleryGPT, which is slightly modified and fine-tuned based on LLaVA architecture leveraging our collected data. We conduct formal analysis generation and zero-shot experiments across several datasets to assess the capacity of our model. The results show remarkable performance improvements comparing with powerful baseline LMMs, demonstrating its superb ability of art analysis and generalization. The codes and model are available at: *https://github.com/steven640pixel/GalleryGPT*.

## CCS CONCEPTS

• **Applied computing** → **Arts and humanities**; • **Computing methodologies** → **Natural language generation**.

## KEYWORDS

Artwork analysis, large multimodal model, dataset

**ACM Reference Format:**
Yi Bin, Wenhao Shi, Yujuan Ding, Zhiqiang Hu, Zheng Wang, Yang Yang, See-Kiong Ng, and Heng Tao Shen. 2024. GalleryGPT: Analyzing Paintings with Large Multimodal Models. In *Proceedings of the 32nd ACM International Conference on Multimedia (MM '24), October 28-November 1, 2024, Melbourne, VIC, AustraliaProceedings of the 32nd ACM International Conference on Multimedia (MM'24), October 28-November 1, 2024, Melbourne, Australia.* ACM, New York, NY, USA, 10 pages. https://doi.org/10.1145/3664647.3681656

*The corresponding author.

## 1 INTRODUCTION

Artwork analysis and composition are integral aspects of art appreciation and creation, often requiring a deep understanding of artistic techniques, styles, and historical contexts. In the past decades, AI systems have been evolving rapidly and demonstrating remarkable success in many fields, even surpass humans [2, 8, 33]. However, it still cannot understand and analyze an artwork like humans since it involves very high-level joint understanding of culture, symbolism, abstractionism, and other aesthetics knowledge, beyond the basic semantic understanding of objects, attributes and relations in natural image understanding. Motivated by the superior ability of deep learning, researchers have employed several advanced techniques, such as convolutional neural networks (CNN) [27, 31], recurrent neural networks [21], and Transformer [4, 55] in style classification, object detection, multimodal retrieval, art visual question answering, and artwork captioning [3, 37, 49]. Despite these advancements,

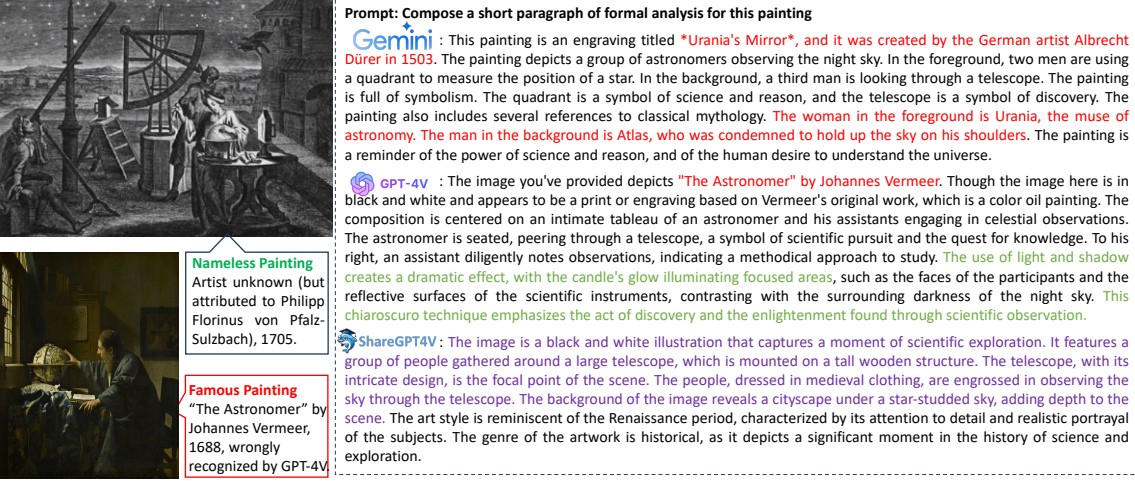

**Figure 1: An example of existing LMMs for art analysis. The painting (a nameless painting to avoid the knowledge memorized in LLMs) on the left-top corner is the target image to be analyzed, and the bottom one, "The Astronomer" by Johannes Vermeer, is wrongly recognized by GPT-4V. The red texts indicate wrong analysis, mainly belong to wrong recognition by Gemini and GPT-4V. The green texts are good parts for "formal analysis". The purple texts denote the factual description of the painting content, which is more similar with the image captioning task.**

there is still a lack of research on composing comprehensive and in-depth analysis for artworks, limited by the model capacity of visual perception and language generation.

The emergence of large foundation models, such large language models (LLMs) [6, 15, 53, 54] and large multimodal models (LMMs) [8, 33, 34, 50], has facilitated the progress across numerous research areas, including text summarization, open-ended question answering, and long-context reasoning, and led to significant advancements [15, 16, 40]. This also makes it possible to enable machine to perceive and understand visual content and generate detail descriptions, such as visual storytelling [23]. However, despite the achievements on natural image understanding, we observe that existing LMMs still cannot comprehend the high-level concepts and generate comprehensive analysis for artworks. As the example illustrated in Figure 1, we test several powerful LMMs, i.e., GPT-4V [26], Gemini [24], and ShareGPT4V [10], with a nameless painting[1] by asking them to compose a paragraph of formal analysis. From the outputs, we can observe that the GPT-4V and Gemini wrongly recognized this painting to another, and then give the analysis based on the knowledge recalled from their language-part memories, which means they function as LLMs in this part. We call this phenomenon as "***LLM-biased visual hallucination***". This phenomenon suggests that in this task, existing LMMs tend to first recognize the given painting is and then give analysis accordingly, while do not focus on the visual content of the painting at the stage of analysis generation. Such a *recognize-then-analyze* procedure highly relies on the accuracy of the recognition, and will fail when the given painting is unknown. Although we have noticed that GPT-4V tries to analyze this painting based on the visual content (green part

shown analysis shown in Figure 1), it still cannot completely escape from the recognize-then-analyze procedure. Meanwhile, some effort has been devoted to enhance the visual understanding of LMMs. For example, ShareGPT4V [10] contributes a dataset with high-quality image and description pairs to fine-tune the LLaVA model [38]. Despite of the effectiveness, these methods still fail to make professional artistic, aesthetic and technical analysis on artworks or paintings, meaning that existing LMMs so far cannot satisfy the requirements of artwork analysis. They may face particular challenges to get generalized into unknown or unseen subjects, thereby calling for better solutions.

To fill the research gap, this paper focuses on generating *formal analysis* for painting images, relying on the LMMs but pushing them to perceive and comprehend the artistic skills or other professional visual aspects shown in an artwork itself. A comprehensive artwork analysis includes many parts, *e.g.*, introduction, cultural, formal analysis, historical context, interpretation, *etc*. Although most of artwork analysis may be done relying only on the external knowledge or subjective opinions, the "form" of the artwork, including color, composition, line, shape, light and shadow, and other visual aspects , still requires vision-based understanding. Therefore, it is necessary to develop a professional LMM specialized for making formal analysis for artworks.

However, there does not exist any formal analysis dataset. It is also time- and labour-expensive to make annotations because writing a formal analysis for an painting requires professional expertise in art analyzing. Motivated by the aforementioned observed LLMs bias by GPT4 and Gemini, we try to access the knowledge memorized in LLMs to produce analyses for known artworks. Specifically, we first collect about 19k famous painting images and corresponding meta data from the Internet. Then, apply LLMs to provide a paragraph analysis only focusing on visual characteristics based on

---

[1]We choose a nameless painting to avoid the knowledge about the painting has been seen during pre-training and Supervised-Fine-Tuning (SFT), since the knowledge about famous ones is memorized by the LLMs. This nameless painting is downloaded from: https://darksky.org/app/uploads/2015/11/8_Florinus_Astronomy.png

the title and artist of the painting, thereby generating the formal analysis. We also prompt the LLMs to compose the formal analysis from some specific form perspectives, such as composition, color, light and shadow, to enhance the richness and diversity of the data. We finally obtain about 50k formal analyses for the paintings we collect and name this dataset as PaintingForm.

Leveraging the PaintingForm dataset, we present GalleryGPT, a large multi-modal model with a LLaVA architecture fine-tuned based on ShareGPT4V-7B [10]. As ShareGPT4V is boosted for image understanding and performs well in visual description generation, we freeze the parameters in vision encoder to retain its superb visual perception ability. Meanwhile, we add a LoRA component to LLM to learn the specific analyzing patterns of paintings. To evaluate the effectiveness of our GalleryGPT, we conduct zero-shot learning on several classic painting analysis datasets, including AQUA [19], ArtQuest [5], and ArtQuest-Type [5] for art visual question answering, and ArtBench [36] for style classification. The results show that our GalleryGPT outperforms several off-the-shelf and adaptive LMMs, demonstrating the superiority of our collected data and impressive performance of GalleryGPT.

In summary, the contributions of this work are as follows:

- We propose to empower the perception ability of LMMs for subtle and specific visual characteristics of artworks, and introduce the task of generating formal analysis to enable the ability by supervised fine-tuning.
- To support the research, we contribute a large-scale and high-quality dataset PaintingForm, acquired by two powerful LLMs, i.e., GPT-4 and Gemini, based on the learnt knowledge about famous paintings. To avoid the leakage of prior knowledge, we ask the LLMs only focus on visual characteristics and do not mention the title and artist in the formal analysis annotation.
- Leveraging the collected dataset, we devise an advanced large multimodal model for painting formal analysis generation, dubbed GalleryGPT, which employs ShareGPT4V as backbone. We evaluate its ability on several painting analysis tasks, and the results demonstrate the impressive performance of it.

## 2 RELATED WORKS

### 2.1 Large Language Models (LLMs)

In the past decade, language modelling has been evolving rapidly and achieved impressive progress [1, 14, 43, 55]. Transformer [55] employed multiple attention blocks and positional embedding to accelerate the recurrent models and made breakthrough in language models. Delvin et al.[14] designed a bidirectional transfomer to learn the contextual embedding of words and made the beginning the era of pre-training language model (PLM) [32, 39, 45–47]. With the large scale tokens pre-training, deep neural models, especially the large language models (LLMs) [6, 11] based on the Transformer [55] structure, demonstrate superb understanding and generation ability, as well as generalizing to downstream tasks, even without any fine-tuning. LaMDA[52] focused on conversational applications and aims to generate more natural and logically-rich dialogue text. InstructGPT[41] designed an effective fine-tuning method that allows LLMs to operate according to desired instructions, leveraging

Reinforcement Learning from Human Feedback(RLHF). It incorporates humans into the training loop using carefully designed annotation strategies. With the instruction fine-tuning based on large foundation models, LLMs, especially the ChatGPT system [25] of OpenAI, also demonstrate emergent capacity of generation [51, 53]. The LLaMA[53, 54] model released by Meta, with its open-source nature and smaller parameter size, has provided an opportunity for many researchers to participate in large language model research and led to rapid research progress in LLMs.

### 2.2 Large Multimodal Models (LMMs)

The blooming of large language models has attracted a great amount of research attention on vision-language interaction and injecting visual knowledge into LLMs. As a paradigm of visual language modal alignment, CLIP [44] implemented contrastive learning on extensive image-text pairs. Subsequent improvements [33, 34] over CLIP utilized enhanced data strategies with greater data diversity for basic visual tasks. Recent research has increasingly focused on pre-traning alignment and visual instruction tuning on top of the LLMs for more complex tasks such as visual question answering and reasoning. MiniGPT-4 [8] demonstrated capabilities in image-text dialogues by aligning queried visual feature with text and feeding queried embedding to LLM. Other prominent examples including LLaVA [38], Qwen-VL [2], InstructBLIP [13], and ShareGPT4V [10] interacted visual features with LLM using a learnable projector or query embeddings, which focus on utilizing more and high quality pre-training and fine-tuning data to understand complex instructions. mPLUG-Owl [58], Shikra [9], and KOSMOS-2 [42] introduced grounding data types and new modularization training to minimize hallucinations and enhance grounding ability. Despite these advancements, exploration for quality and format of images-instructions highlights a critical area for future large multimodal models improvement.

### 2.3 AI for Art Analysis

With the great success of deep learning in CV and NLP in the past decade, AI for art analysis has also been evolving with rapid progress. Early works most depended on hand-crafted features and explored the classification and recognition problem [7, 20, 28, 30, 48]. With the great success of pre-trained language models (PLM) [29, 57] and visual-language pre-training (VLP) [35, 44], research on artwork analysis also achieves significant progress. CLIP-Art [12] leveraged the image-text pairs of artworks in SemArt [18] to fine-tune the CLIP and gains impressive improvements. Garcia et al. [19] contributed a dataset of art visual question answering, named AQUA, and proposed a knowledge-based VIKING model, which achieves the best performance. Bleidt et al. [5] pointed out there exists language bias hidden in the question-answer pair, which may induce the model to ignore the visual information. To address this issue, they further proposed a strategy to eliminate the bias and introduced an ArtQuest dataset and implemented PrefixLM to learn the patterns between paintings and questions.

Despite these simple tasks investigated in deep learning era, people also expect the AI system could provide comprehensive analysis for artworks, which may benefit the art education and assist human writing the commentary. Recently, LLMs and LMMs have

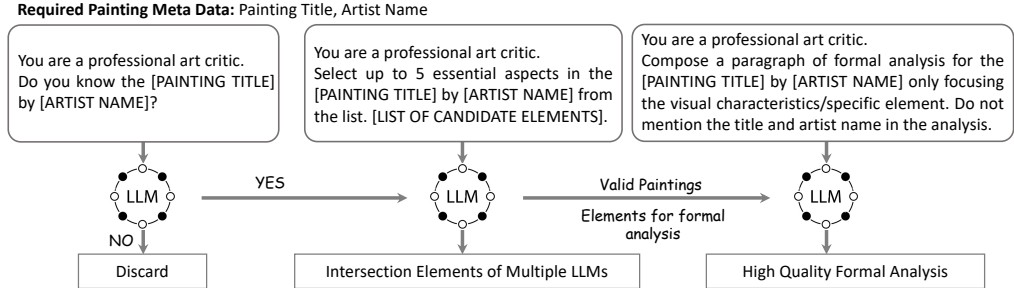

**Figure 2: The overall pipeline of constructing our PaintingForm collection. Our annotation process only depends on the language model, without vision information. The prompt illustrated here just a simple version for exhibition, we actually elaborately designed the prompts.**

been demonstrating superb understanding and generation ability in many areas, which raise the potentials to compose comprehensive analysis for artworks. However, we argue current LMMs might be biased by the knowledge memorized in language and cannot generalized to nameless artworks. In this work, we try to implement formal analysis with LLMs to make them focusing on visual elements in art analysis.

## 3 DATA COLLECTION

### 3.1 Primary Philosophy

A successful artwork analysis, as been discussed before, needs to cover multiple types of content including background introduction, formal analysis, historical context, *etc*, most of which rely on external knowledge and subjective opinions except formal analysis. As we have known, existing of-the-shelf LMMs have been capable of providing factual information and description with relevant knowledge while tend to ignore visual analysis, we want to enhance the existing artwork analysis models to focus more on visual elements. The research purpose in this work is to make the developed model to emphasize on visual comprehension rather than recognizing the artwork and retrieving knowledge from their language memory, in other word, to twist the *recognize-then-analyze* procedure of existing LMMs on artwork analysis which may cause "*LLM-biased visual hallucination*".

To support the development of such a professional LMM specialized for artwork analysis, we try to construct a large-scale artwork analysis dataset, including images and corresponding analysis, so that we can fine-tune the pre-trained LMMs. More importantly, to push the LMMs to focus on visual comprehending of the artworks through fine-tuning, we need high-quality formal analysis annotations in our dataset. However, manually annotating artworks with comprehensive analysis requires professional expertise in artwork analyzing and it is hard and expensive to recruit so many professional art critics. Besides, even with experts, it is still time-consuming to annotate large number of artworks.

Based on these motivations and challenges, we collect a painting dataset, named PaintingForm, consisting of paintings and associated formal analyses annotations for each painting. Since manual labeling is affordably expensive in terms of both timing and economic cost, we elaborately leverage the powerful LLMs to produce formal analysis on famous paintings, ensuring they have enough

knowledge to support them producing high-quality analysis. The overall pipeline of our data collection process is shown in Figure 2 and more details are described in subsequent sections.

### 3.2 Painting Source

With the development of digital technology in recent years, large collections of artworks have been digitized and stored, and easily to accessed via the Internet. We focus on famous paintings and choose 1st Art Gallery[2] as our painting source. To make the LMMs, *e.g.*, GPT-4 and Gemini, able to provide accurate and comprehensive formal analyses, we choose the paintings of *500 Most Popular Paintings*[3] and *Most Popular Artists*[4] to ensure the LLMs knowing the paintings. With such condition, we obtain 19,295 paintings (including 18,795 of most popular artists). To ensure the LLMs know the paintings, we first ask Gemini to answer if it knows the painting with title and artist name. We also filter out some paintings without certain title and annotated as "unknown", and obtain 18,526 paintings in the end. We select 5000 less popular paintings for test and reserve 13,526 for training, test samples of which are identified by the wishlist count from source website, and the distribution across artists is also considered. We illustrate the statistics of artist distribution of the Most Popular Artists[5] in Figure 3. From the statistics, we observe that the dataset includes most paintings of Vincent Van Gogh, resulting in 1458 retained and 225 filtered, and Jacques Louis David with the fewest paintings included in the dataset, about 100 paintings. We have also verified that all the paintings are free to use without copyright concerns, as provided by the website[6].

### 3.3 Formal Analysis Annotation

As aforementioned, we employ powerful LLMs to generate high quality formal analysis, after ensuring the paintings are known by the LLMs. Note that we only provide the LLMs the title and artist

---

[2]https://www.1st-art-gallery.com/

[3]https://www.1st-art-gallery.com/most-popular-paintings.html

[4]https://www.1st-art-gallery.com/browse-by-artist-a-z.html. Note that here we only use the most popular artists, not all, resulting in 48 artists in total.

[5]We do not include the 500 most popular paintings in this statistics because about 150 artists only contribute one painting in the whole gallery.

[6]https://www.1st-art-gallery.com/copyrights.html, The copyright contents are as [Accessed on 4 April, 2024.]: "All images on our site are either licensed or in public domain because their copyright has expired. This applies to the United States, Canada, the European Union and the countries with a copyright term of life of the author plus 70 years."

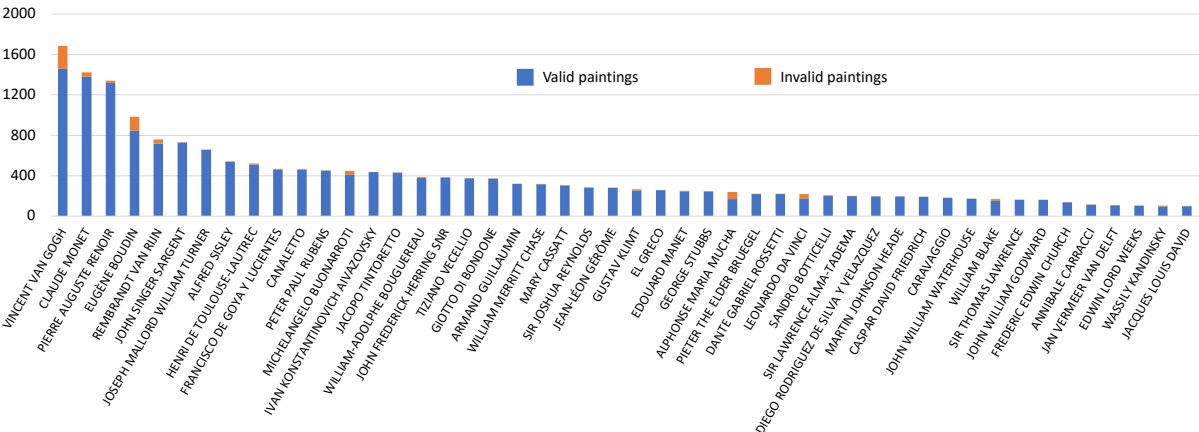

**Figure 3: An statistic illustration of the distribution of paintings by each artist in *Most Popular Artists*. Valid paintings denote the reserved paintings after the filtering rules in Section 3.2, and the invalid paintings are the discarded ones. We finally filter 769 and reserve 18,026 paintings for this part (not include the 500 most popular paintings). We do not include the 500 paintings for illustration because: 1) all the 500 popular paintings are reserved, and 2) there exist about 150 artists have only one paintings in this gallery, resulting in severe long-tail distribution here.**

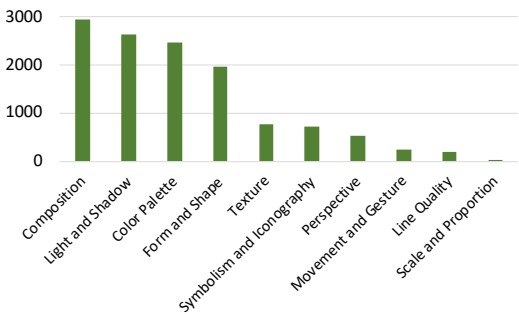

**Figure 4: Statistics of elements of formal analysis. Only very few paintings (8 in total) contain the elements *Scale and Proportion*.**

name of the paintings, and do not input any visual information, *i.e.*, the painting image, to the LLMs. Specifically, we employ two powerful LLMs, GPT-4 and Gemini in this work, to retrieve the learnt knowledge with the title and artist name of a certain painting and generate a paragraph of formal analysis only focusing on visual characteristics. To make the formal analyses more diverse, we ask the LLMs to generate two-level formal analyses: 1) overall formal analysis, and 2) formal analysis from a certain perspective, *e.g.*, color, composition, and *etc*. To avoid the "LLM-biased visual hallu-cination", we ask the LLMs do not mention the title and artist name in the generated formal analysis. In other words, one cannot easily identify the corresponding painting solely depending on a specific formal analysis. For the perspective specified formal analysis an-notation, we first ask GPT-4 and Gemini to provide up to 5 most important perspectives of the given painting independently, and utilize the intersection of GPT-4 and Gemini predictions as the final perspectives. The statistics of perspectives is shown in Figure 4, from which we observe that *Composition, Light and Shadow, Color Palette, Form and Shape* are the most common perspectives of the

paintings. Finally, based on the selected perspectives, we employ LLMs to annotate the formal analysis only focusing on a specific perspectives similar with the overall setting.

## 4 THE PROPOSED GALLERYGPT

To verify if our proposed PaintingForm dataset could empower LMMs more superior painting analyzing ability, we develop a large multimodal model, GalleryGPT, based on ShareGPT4V-7B that has demonstrated impressive performance across several multimodal tasks. We conduct supervised fine-tuning (SFT) on our Painting-Form dataset to enable the LLMs to analyze paintings focusing on visual elements.

### 4.1 Architecture

As pointed out before, our goal is enabling LMMs to analyze art-works focusing more on visual elements, rather to design a new architecture or model. Therefore, we introduce the GalleryGPT em-ploying ShareGPT4V-7B as backbone, which follows the LLaVA [38] architecture and consists of three components: 1) The visual en-coder, which is a vision transformer (ViT) [17] borrowed from the CLIP-Large [44]. Similar with CLIP-Large, the visual encoder takes 336*336 shape as input size and divides it into 14 patches, resulting in 576 input tokens; 2) The projector, a two-layer MLP to project the visual representation into the language semantic space, LLMs space in specific; 3) The LLM, which is based on Vicuna-v1.5 [51] and LLaMA2 [53], employing the decoder-only architecture. For the whole setting, we follow ShareGPT4V-7B focusing on the 7B model. Besides, to keep the superior visual perception and describ-ing ability of ShareGPT4V-7B, we slightly modify the LLM in our GalleryGPT. Specifically, we add several LoRA [22] modules to learn the formal analysis specific patterns and freeze the LLM in ShareGPT4V-7B to keep its superb content describing ability.

## 4.2 Supervised Fine-Tuning

Numbers of previous works [26, 41, 51, 53, 56] have demonstrated that based on billions or trillions of tokens pre-training and elaborate supervised fine-tuning, the LLMs exhibit creative emergent ability and is able to generalize to multiple tasks without further training. Following this inspiration, we implement supervised fine-tuning (SFT) with our LLM-generated formal analyses, paired with the corresponding painting. During SFT stage, we set the learning rate as 2e-5, batch size as 16, and fine-tune 10k steps on an A100 GPU with 80G memory. For the LoRA module, we set the lower rank as 128 and the alpha is 256. Since our analysis texts do not contain any identifying information of the paintings, the LMMs cannot retrieve the learnt knowledge to associate with the paintings. Instead, with our SFT optimization, the LMM tries to perceive the subtle elements and art skills presented in the painting, and matches them with the corresponding descriptions in the formal analysis. Through such fine-tuning, our GalleryGPT could be empowered artwork analyzing ability and generalized to other art analyzing tasks, *e.g.*, style classification.

## 5 EXPERIMENTS

### 5.1 Formal Analysis Generation

Since our data and optimization goal are for painting formal analysis generation, we first directly evaluate the quality of generated formal analyses on our reserved test set, with 5000 paintings. We also test with several popular and powerful open-source LMMs, including LLaVA-1.5 [38], Qwen-VL-Chat [2], and ShareGPT4V[7] [10], and all the LLMs employ the same prompt: "*Please compose a coherent paragraph of formal analysis focusing on visual characteristics*". We combine the formal analyses annotated by GPT-4 and Gemini for each painting to formulate the ground truth descriptions. Finally, we employ image captioning metrics to evaluate the generated formal analyses because formal analysis is also a kind of description. As the evaluation results shown in Table 1, we observe that our GalleryGPT outperforms all the other LMMs with a remarkable improvement. Among all the baseline LLMs, LLaVA-1.5 performs the worst and is significantly lower than others, while Qwen-VL and ShareGPT4V achieve similar performance. From these observations, we can conclude that, benefiting from the SFT with PaintingForm, the GalleryGPT achieves impressive improvements comparing with ShareGPT4V-7B, which mainly focuses on natural image-text pairs (only a few art data). This verifies the effectiveness of our high quality art analysis data collections.

### 5.2 Generalizing to Other Art Analysis Tasks

LLMs and LMMs have demonstrated superior generalizing ability to many downstream tasks. To verify the generalization of our GalleryGPT, we conduct visual question answering and style classification for paintings on several existing datasets. The details of the dataset are as following:

- **AQUA** [19]: is an Art Question Answering dataset based on SemArt [18]. The question-answer pairs are generated

---

7Here we implement 7B model for fair comparison for all baseline and omit 7B in the table. Note that in [10], ShareGPT4V indicates the dataset and ShareGPT4V-7B is the model, but in this table we omit 7B for unifying all the model names.

**Table 1: Performance comparison of formal analysis generation measured by the captioning metrics, evaluated on our test set with 5000 paintings.**

| Model | BLEU | GLEU | METEOR | ROUGE |
|---|---|---|---|---|
| LLaVA-1.5 [38] | 9.87 | 14.59 | 26.19 | 26.37 |
| Qwen-VL-Chat [2] | 13.65 | 16.42 | 29.78 | 26.72 |
| ShareGPT4V [10] | 12.38 | 16.14 | 31.53 | 26.63 |
| GalleryGPT (ours) | **21.23** | **21.68** | **37.62** | **31.34** |

**Table 2: Zero-shot performance comparison on several artwork analysis tasks, including question answering and classification. SoTA-1 and SoTA-2 denote previous state of the arts without and with PLM (including fine-tuning on corresponding datasets). Both SoTA-1 and SoTA-2 denote different methods for different tasks.**

| Model | AQUA | ArtQuest | ArtQuest-Type | ArtBench |
|---|---|---|---|---|
| SoTA-1 | 22.40 | 2.40 | - | - |
| SoTA-2 | 55.50 | 50.2 | 81.8 | - |
| LLaVA-1.5 | 22.13 | 8.66 | 34.82 | 28.1 |
| Qwen-VL-Chat | 18.67 | 7.86 | 26.29 | 28.0 |
| ShareGPT4V | 23.62 | 7.91 | 38.07 | 31.7 |
| GalleryGPT (ours) | **24.08** | **9.51** | **43.94** | **34.0** |

by powerful question generation models based on paintings and comments provided in SemArt. The dataset contains 29,568, 1,507, and 1,270 samples in training, validation, and test splits, respectively.

- **ArtQuest** [5]: is the debiased version of AQUA, including 6414 test cases, which mainly focuses on eliminating the bias hidden in language. With such debias operation, most question cannot be answered without visual content.
- **ArtQuest-Type** [5]: is simple setting of ArtQuest, which requires the model only answer the type of the painting. The test set contains 1069 samples.
- **ArtBench** [36]: is originally for Artwork generation. We random sample 1000 paintings from the test set (10000 paintings in total), and use the provided style labels for classification.

We exhibit the results in Table 2, from which we can see that our GalleryGPT also significantly outperforms all the baseline LMMs and demonstrates its generalizing ability for downstream art analysis tasks. An interesting observation is that LLaVA-1.5 performs much better than Qwen-VL-Chat, showing an opposite result to the task of formal analysis generation. We have checked and analyzed the results, and hypothesize that may come from the output format mismatching. Since all the three QA datasets follow the open-ended answering setting, and then extract the matching strings from the outputs. Another piece of evidence is that Qwen-VL-Chat achieves similar performance to LLaVA-1.5 when the output strictly adheres to the given labels.

To comprehensively analyze the ability of our GalleryGPT, we also compare it with several state of the arts and show the results in the top block in Table 2, which are specific trained on the corresponding datasets. SoTA-1 denotes the models without pre-training

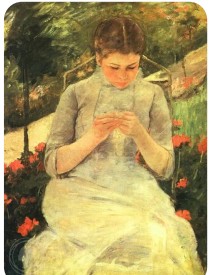

**Prompt:** *Please write a paragraph of formal analysis for this painting.*

**LLaVA-1.5-7B**

The image features a woman sitting on a chair, wearing a white dress and holding a needle in her hand. She appears to be engaged in a sewing activity, possibly mending her clothing. The scene is set in a garden, with a potted plant located near the woman. The woman's attire, including the white dress, and the garden setting create a sense of tranquility and simplicity. The focus on the woman's hands and the needle she holds emphasizes her concentration on her task, while the potted plant adds a touch of nature to the scene. Overall, the image captures a moment of quiet contemplation and creativity in a serene outdoor setting.

**ShareGPT4V-7B**

The image is a captivating impressionist painting that captures a moment of tranquility in a garden. The central figure is a young woman, elegantly dressed in a white dress, seated on a bench. She is engrossed in her hands, perhaps lost in thought or simply enjoying the serenity of her surroundings. The garden around her is a riot of colors, with red flowers adding a vibrant contrast to the lush greenery. The painting is executed in an impressionist style, characterized by loose brushstrokes that capture the interplay of light and color. The artist's use of color and light brings a sense of depth and realism to the scene, while the focus on the woman's contemplative state adds a layer of emotional depth to the composition. The overall effect is a harmonious blend of color, light, and emotion, a hallmark of the impressionist genre.

**GPT-4V**

This painting depicts a young woman absorbed in her needlework, seated amidst a lush, verdant garden setting. The artist has employed a vibrant color palette, with the warm tones of the woman's dress and the surrounding foliage creating a harmonious and inviting atmosphere. The woman's concentration is evident in her downcast gaze and the delicate movements of her hands, suggesting a sense of tranquility and introspection. The interplay of light and shadow across the scene further enhances the painting's sense of depth and atmosphere, drawing the viewer into the intimate, contemplative moment being captured. Overall, the work conveys a serene, introspective mood, highlighting the beauty and quiet simplicity of the everyday.

**GalleryGPT (Ours)**

This exquisite piece of artistry is a testament to the mastery of capturing human emotion and intimate moments. The visual characteristics of the painting are striking, with the use of soft, muted colors that create a sense of warmth and tranquility. The central figure, a young woman, is depicted in a domestic setting, engrossed in the task of sewing. The artist's use of light and shadow is particularly noteworthy, with the light source seemingly emanating from the left, casting gentle shadows that add depth and dimension to the scene. The woman's facial expression and body language convey a sense of concentration and introspection, while her surroundings, including the ornate chair and the patterned fabric, add a layer of visual interest. The artist's brushwork is delicate and precise, particularly in the rendering of the woman's hands and face, which are imbued with a sense of realism and detail. The overall composition is balanced and harmonious, with the figure and her surroundings occupying the majority of the canvas, leaving little space for distraction. This, coupled with the artist's use of perspective, draws the viewer's attention towards the figure, making her the focal point of the painting.

**Figure 5: An example for qualitative comparison of formal analysis generation by several powerful LMMs. *Purple* texts denote the factual content description, and the *Blue* texts are for formal analysis. The formal analysis generated by our GalleryGPT covers more visual elements, *e.g.*, color, light and shadow, depth, composition, and perspective, than other LLMs, even the powerful GPT-4V.**

language model (PLM), *e.g.*, BERT, and SoTA-2 denotes the models leveraging PLM features or fine-tuning based on PLMs. From the results we can see, almost all the current LMMs only achieve the performance around SoTA-1 on AQUA, *i.e.*, specifically training on this dataset, while they can significantly outperform SoTA-1 on ArtQuest. This reason may come from the language bias hidden in AQUA dataset [5]. For all the dataset, the performances of LMMs are far from the SoTA-2, which means the generalization ability of LMMs on art analysis is still weak and there exists a large space for them to get improved. Our hypothesis on such results is that artworks may contain high-level and abstract concepts to be perceived more subtly, which also motivates us to make further endeavour on this in the future work.

## 5.3 Multimodal Benchmarks

Since GalleryGPT is implemented based on ShareGPT4V-7B, it inherently remains an LMM. Therefore, we also conduct experiments on several LLM benchmarks and compare it with baseline LLMs[8] to evaluate the multimodal understanding ability. As the comparison results illustrated in Table 3, we can see our GalleryGPT exhibits comparable performance with ShareGPT4V, *i.e.*, achieving improvements and reducements in slight fluctuations. In other words, our GalleryGPT, further fine-tuned with elaborately collected painting-analysis pairs, not only achieves better painting analysis performance, including formal analysis generation (Table 1) and other downstream tasks (Table 2), and also retains its superior multimodal understanding ability. These observations further verify the

**Table 3: Comparisons with powerful baselines on several LMM Benchmarks. The best and 2nd-best results are in bold and underlined, respectively. All the results of baselines are referred from [10].**

| Model | MMB | LLaVA-W | MM-Vet | SQA |
|---|---|---|---|---|
| LLaVA-1.5 | 64.3 | 63.4 | 30.5 | 66.8 |
| Qwen-VL-Chat | 60.6 | - | - | 68.2 |
| ShareGPT4V | 68.8 | **72.6** | **37.6** | 68.4 |
| GalleryGPT (ours) | **70.8** | 69.9 | 35.5 | **69.1** |

effectiveness and contributions of our collected PaintingForm and developed GalleryGPT.

## 5.4 Qualitative Analyses

To visually and straightforwardly compare our GalleryGPT with other LMMs, inncluding LLaVA-1.5, ShareGPT4V-7B, and GPT-4V, we test some unpopular paintings in our test split. As illustrated in Figure 5, we observe that LLaVA tends to describe the factual content (in purple color) presented in the painting, and fails to analyze the paintings from the perspective of art criticism. ShareGPT4V-7B, the backbone of our model, also pays more attention on the content description, because the SFT data it utilized for fine-tuning is focusing on describing the images in detail. Thanks to the small subset of art data in ShareGPT4V, it also exhibits some capacity for art analysis, albeit limited to general analyses and overlooking the subtle visual characteristics. GPT4V, as the most famous and capable LMM at current stage, demonstrates much better analyzing ability on paintings in terms of their form, while still fails to capture some subtle artistic aspects, such as composition and

---

[8]Instead of listing all the baselines in [10], here we only list the superior and representative ones for comparison.

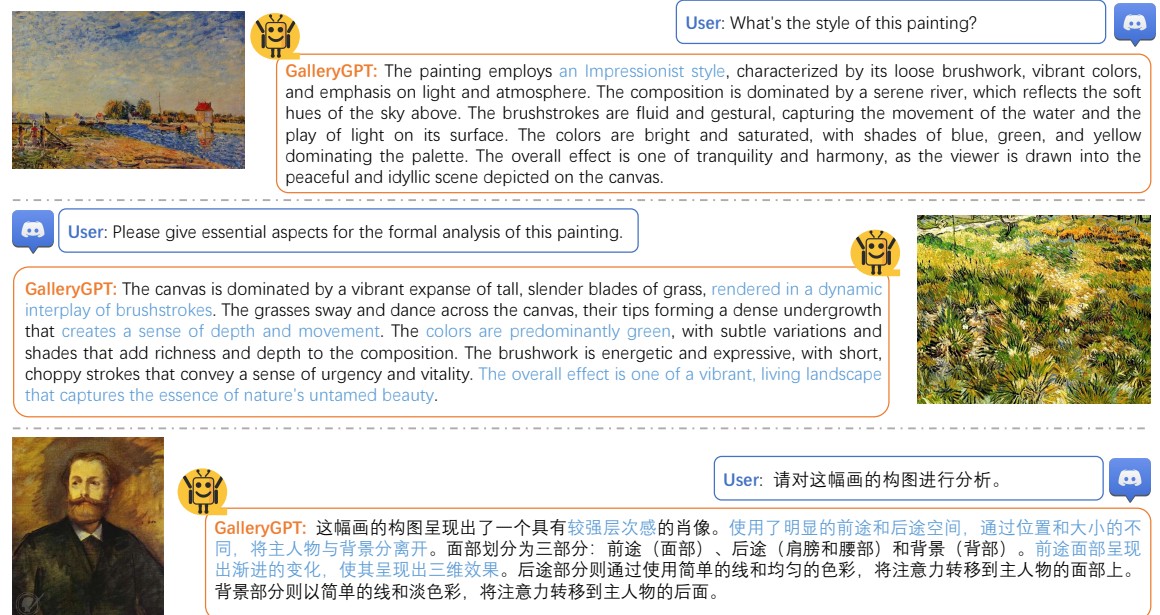

**Figure 6: Multimodal dialogue examples with our GalleryGPT. *Blue* texts denote high quality formal analysis. We also test dialoguing in Chinese, even we do not fine-tune GalleryGPT with Chinese painting-analysis pairs. The English translation of Chinese conversation can be found in Supplementary.**

depth. Obviously, our GalleryGPT demonstrates superb capability of comprehensively analyzing the artworks, which not only briefly describe the factual content of the painting, but also focuses more attention on analyzing the subtle artistic elements, including color, light and shadow, depth, composition, and perspective. These observations have definitely verified the superiority of our GalleryGPT for artwork analyzing.

We also investigate the dialogue capability of our GalleryGPT. We show several dialogue examples in Figure 6. The examples demonstrate that our GalleryGPT is able to follow the user intention in the conversation. As shown in the second case, for instance, we ask it to give "essential aspects" for formal analysis, it just provides us a brief analysis with essential content, which is much shorter than the one provided in Figure 5. Besides, we further explore the multilingual capabilities of GalleryGPT, even though we have not provided any multilingual painting analyses for supervised fine-tuning[9]. As shown at the bottom in Figure 6, we chat with GalleryGPT in Chinese[10], it still generates high quality answer focusing on "composition" and exhibits strong multilingual ability.

In summary, these qualitative comparisons and dialogue examples have verified the superior art analysis ability of our GalleryGPT, as well as illustrating the quality of our PaintingForm dataset. For more examples of this part, qualitative comparison and chat conversation, could be found in Supplementary.

---

[9]Actually, the foundation models, LLaMA and LLaVA, are pre-trained and fine-tuned with several multilingual corpuses.
[10]A reviewer pointed out that the "前途" and "后途" are typos and should be "前景" and "后景", since this analysis is generated by our GalleryGPT and we have not modified it. As the English translation in Figure. 6 in Supplementary, the meanings are the same with the suggestions of the reviewer. We sincerely appreciate the reviewer providing such careful and helpful comments and suggestions.

## 6 CONCLUSIONS AND FUTURE WORKS

In this work, we targeted at artwork analyzing with large multimodal models (LLMs). We first tested several LLMs and pointed out that current LLMs may suffering from "LLM-biased visual hallucination" issue, resulting in weak generalization ability to nameless paintings. To make the LMMs to analyze artworks focusing on visual elements and easy to generalize, we proposed to conduct SFT of LMMs with formal analysis. To support this research, we first elaborately designed an LLM-based data collection pipeline to construct high quality painting-analysis pairs. We also employed ShareGPT4V to implement SFT on the collected data, derived our GalleryGPT. We conducted extensive experiments to verify the effectiveness regarding to formal analysis generation, generalizing to down stream tasks, and LMM benchmarking. The results demonstrated the effectiveness of the collected data and introduced GalleryGPT.

For future works, on one hand, we will step further to investigate the generalization issue mentioned in Section 5.2 to explore the ability of LMMs in art analyzing. For example, devising a ChatGPT-like art assistant to help more people appreciate or learn to appreciate artworks. On the other hand, we only collect paintings in this work, while artworks consist of multiple types, *e.g.*, ceramics (paintings on a curved surface) and sculptural in 3D. Therefore, we will try to make the research more widely and empower LMMs to assist human in artwork analyzing, *e.g.*, drafting formal analysis, classifying the unseen artworks, *etc*. We also hope that our work can inspire more researchers to explore AI, especially for LMMs, within the field of art analysis.

## ACKNOWLEDGMENTS

This work is supported by the National Natural Science Foundation of China under grant 62102070, 62220106008, and 62306065. This research/project is supported by the National Research Foundation, Singapore under its Industry Alignment Fund – Pre-positioning (IAF-PP) Funding Initiative. Any opinions, findings and conclusions or recommendations expressed in this material are those of the author(s) and do not reflect the views of National Research Foundation, Singapore.

We also sincerely thank all the ACs and reviewers for their efforts on our work and appreciate the useful comments for improving it.

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
