# OpenReview forum: "GalleryGPT: Analyzing Paintings with Large Multimodal Models"
_acmmm.org/ACMMM/2024/Conference — MM2024 Oral_

### Official Review · Reviewer_mgdk · 2024-05-24

**Rating:** 6
**Confidence:** 4

**Summary:**

This paper outlines the core challenges of using LLM for artistic analysis, specifically the issue of "LLM-biased visual hallucination". The authors address this by collecting a dataset and employing fine-tuning techniques to focus the attention of the LLM on visual characteristics, moving away from the "recognize-then-analyze" procedure. The structure of the paper is coherent, with sound methodology and thorough experimentation.

**Strengths:**

- The writing is clear and comprehensible. The authors present their narrative in a logical, fluid, concise, and accurate manner, which is excellent.
- The problem addressed is innovative. The authors incisively identify a critical issue in existing LLM techniques within the field of art analysis, "LLM-biased visual hallucination," which is of significant importance to the domain.
- Both the methods and evaluations are very solid.

**Limitations:**

**related work**
- As the authors have mentioned, in the field of art analysis, the definition of artwork extends beyond painting, encompassing sculptures, ceramics, and more, which can all be utilized for visual characteristics description. Taking ceramics as an example, it can be understood as painting on a curved surface. On a two-dimensional level, patterns and colors can be described, while on a three-dimensional level, the shape, form, and manufacturing process can also be addressed. There is existing work on generating captions or extended captions (descriptions) for these pieces. The authors might consider discussing the potential integration of these works with LLM in either the related work or future work sections in subsequent updates. Of course, if time permits, conducting some preliminary tests could make the article more comprehensive, depending on the authors' time constraints. I will provide some papers for your reference.
1. Fang Liu, Mohan Zhang, et al. Feature fusion via multi-target learning for ancient artwork captioning. Information Fusion, 2023.
2. Shurong Sheng and Marie-Francine Moens. Generating Captions for Images of Ancient Artworks. ACM MM, 2019.

**Typos**
- There are some minor errors in the Chinese text in Figure 8. '前途' and '后途' should be corrected to '前景' and '后景,' respectively. '前途' is commonly used to describe the future prospects of a person's career or work. While '前景' is synonymous with '前途,' it also carries the additional meaning of describing the part of a painting or stage scene that is closer to the observer. In this case, '层次感' refers to the depiction of spatial depth between foreground and background. For instance, the foreground is usually portrayed clearly, while the background tends to be blurred, aligning with the actual visual experience of humans.

**Others advise**
- This topic makes me immediately think of whether such technology could be combined with the description (the purple part in Figure 1) to assist in the development of assistive technology for museums, art galleries or online scenarios? I think this could be an interesting direction.

**Overall**
- There is no doubt that this article provides new insights into the field of art analysis. I highly recommend this article for publication in the MULTIMEDIA conference. The authors could have gone further to make this article even better based on the reviewers' comments.

**Suitability:**

3

---

### Official Review · Reviewer_NMkE · 2024-06-03

**Rating:** 3
**Confidence:** 4

**Summary:**

The paper introduces the task of “Formal analysis generation” for paintings with large multimodal models (LMMs). In pursuit of this, the paper shows a technique to collection of a dataset based on the definition of 'formal analysis' and leveraging the capability of LLMs and LMMs to generate a formal label for the set of images. Then they fine-tune a LMM on this data using LoRAs for language component calling the final model - GalleryGPT. Additionally, the paper analyzes the performance of GalleryGPT with zero-shot experiments for demonstrating the capacity of the fine-tuned multimodal model.

**Strengths:**

1. It is a good idea to use openly available reproductions of paintings for formal analysis using SOTA LLMs and LMMs. This avoids copyrights issues.

2. GalleryGPT shows good zero-shot downstream performance on various other datasets (Tab. 2) .

3. Retention of multi-lingual ability.

4. GalleryGPT is shown to be better than its relevant counterparts.

**Limitations:**

1. The proposal lacks rigor. The entire pipeline seems like any ML project: Create a relevant dataset (Paintings-Form) for a specific task (formal analysis), and fine-tune it with a SOTA architecture (ShareGPT). The paper would become stronger if it dives deeper into the idea of formal analysis and how it differs from regular image analysis, empirically through experiments or even some basic surveys with human experts. I don’t know which experiments would that be though. I can just imagine the direction, not specific steps.

2. Formal Analysis Annotation: The chosen models (ChatGPT, Gemini) are arguably one of the best models out there. However, the ‘best’ refers to their performance on a range of tasks. Therefore, the choice of using only these is unclear. There are other fairly good (and openly available) models like llama3, mistral, phi-2, etc whose capabilities could be combined to get a more generic formal labels.

3. Metrics / performance comparison: Often LLMs and LMMs show improved performance with better prompting strategies. This analysis is missing in the paper: For eg: In the prompt, define what formal analysis mean and which elements to focus on and then see the results of the generation. I tried a few examples with this and it seems to me that models like ShareGPT4V and LLaVA-1.5-7B are good in considering the definition of the formal analysis and integrating it into the results. An empirical study of this can easily show whether SFT is really helpful, therefore showing whether GalleryGPT is worth the effort. [a prompt similar to stage 2 - shown in section 2 of supplementary]

4. Title: The title should reflect what the paper proposes - formal analysis, not generic ‘painting analysis‘

5. After the fine-tuning (SFT), does GalleryGPT still retain the capacity for image analysis? Although not so important, but good to know if it retains the capability for image-analysis or is overfitted on the Paintings-Form dataset.

**Formatting Comments:**

1. In line 469-470, it is mentioned that the authors ask Gemini if it knows the painting with the title, however in Fig. 2 the icon-symbol is of ChatGPT. Since the proposal uses Gemini and ChatGPT, it should be made clear.

2. (follow-up of the above point) Fig. 2 uses an icon-symbol to represent LLMs in general or only refers to ChatGPT? It would be better to not use that symbol, since it misrepresents the article’s claim.

**Suitability:**

3

---

### Official Review · Reviewer_6mAc · 2024-06-09

**Rating:** 4
**Confidence:** 3

**Summary:**

This paper proposed GalleryGPT, a LLM that customized for artwork analysis. In particular, the authors constructed a dataset with existing artworks and build their corresponding caption pairs with LLMs, suchas GPT and Gemini. Next, the authors finetuned LLaVA to improve its performance on artwork analysis. Extensive experiments have demonstrated the effectiveness of their method.

**Strengths:**

1. The problem is interesting. Existing LLMs lack knowledge or expert skills for artwork analysis. Thus, by constructing a new dataset in this domain, we can build a more powerful multi-modal LLM to handle this work.

2. Experiments are quite comprehensive. The proposed GalleryGPT has achieved clear advantage over existing closed-source models. It provides a practical value.

3. This paper is easy to follow. Figures are very clear.

**Limitations:**

1. My primary concern is that GalleryGPT appears more as an application of multi-modal LLMs in a practical scenario rather than introducing new techniques. The main contribution seems to be the new dataset, PaintingForm, and the finetuning of LLaVA on this dataset, which does not present novel methods.

2. Furthermore, for readers not familiar with artwork analysis, it may be challenging to understand the significance of GalleryGPT's outputs, especially in comparison with other models. Unlike conventional hallucinations, such as clearly missing objects, the differences in artwork analysis can be subtle. The metrics in Table 1 are typical NLP metrics, which might not fully capture the depth and quality of analyses from an art historical or critical perspective.

**Suitability:**

2

---

### Meta-Review · Area_Chair_g4Hp · 2024-07-09

**Recommendation:** Accept (Oral)
**Confidence:** 5

**Metareview:**

In this paper an LLM specifically targeting Art Analysis is proposed. The reviewers had some initial doubt but the rebuttal took away most of the concerns and now there are two borderline accepts and one accept. Some questions remain on the technological innovation in general but the reviewers clearly see the value for the art domain and sufficiently interesting technology. Reviewers also indicate that working with people having expertise in art would be beneficial.